# Biochemical and Histological Differences between Longitudinal and Vertical Fibres of Dupuytren’s Palmar Aponeurosis and Innovative Clinical Implications

**DOI:** 10.3390/ijms25136865

**Published:** 2024-06-22

**Authors:** Caterina Fede, Ludovico Coldebella, Lucia Petrelli, Franco Bassetto, Cesare Tiengo, Carla Stecco

**Affiliations:** 1Department of Neurosciences, Institute of Human Anatomy, University of Padova, Via Aristide Gabelli 65, 35127 Padova, Italy; lucia.petrelli@unipd.it (L.P.); carla.stecco@unipd.it (C.S.); 2Plastic and Reconstructive Surgery Unit, University of Padova, Via Nicolò Giustiniani 2, 35128 Padova, Italy; ludovico.gss@gmail.com (L.C.); franco.bassetto@unipd.it (F.B.); cesare.tiengo@unipd.it (C.T.)

**Keywords:** palmar fascia, Dupuytren’s disease, collagen, fibrosis, inflammation

## Abstract

Dupuytren’s disease, a chronic and progressive fibroproliferative lesion of the hand, which affects the palmar fascia, has a recurrence rate after selective aponeurotomy of 20–40% at 5 years. This study focused, for the first time, on the microanatomical and histopathological characteristics of the longitudinal and vertical fibres (usually spared during surgery) in the aponeurosis with Dupuytren’s disease, in different stages of the Tubiana’s classification. Twelve human samples were collected and analysed by immunostaining, Total Collagen Assay, ELISA Immunoassay, and immunoblotting for the Von Willebrand factor, α-Sma, D2-40, CD-68, Total Collagen, Collagen-I and III, IL1β, TNF-α to analyse the blood and lymphatic vascularization, the amount and distribution of collagen, and the inflammation. The results show a progressive increase in the arterial vascularization in the vertical fibres (from 8.8/mm^2^ in the early stage to 21.4/mm^2^ in stage 3/4), and a parallel progressive decrease in the lymphatic drainage (from 6.2/mm^2^ to 2.8/mm^2^), correlated with a local inflammatory context (increase in IL-1β and TNF-α until the stage 2) in both the longitudinal and vertical fibres. The acute inflammation after stage 2 decreased, in favour of a fibrotic action, with the clear synthesis of new collagen (up to ~83 µg/mg), especially Collagen-I. These results clearly demonstrate the involvement of the septa of Legueu and Juvara in the disease pathology and the modifications with the disease’s progression. A greater understanding of the pathology becomes fundamental for staging and the adequate therapeutic timing, to obtain the best morpho-functional result and the lowest risk of complications.

## 1. Introduction

The palmar fascia is a three-dimensional anatomical structure that plays an essential role in hand physical function. The palmar aponeurosis is composed of a rich three-dimensional network of longitudinal, transverse, and vertical bundles arranged orthogonally to each other. The first run distally thus producing longitudinal bundles, which travel to the fingers; the transverse fibres are the deepest layer of the palmar fascia, in a band that is approximately 2 cm wide, proximally to the distal palmar crease; and the vertical fibres connect the palmar aponeurosis to the thenar and hypothenar eminences [1,2,3]. The vessels, nerves, and receptor systems are located inside this 3D network.

The palmar aponeurosis has, for years, been considered functionally useless except as a protective fibrous layer, but over time, it has been increasingly understood it takes on an anchoring function to the skin and metacarpal periosteum, and it acts as a guide channel for the tendons, muscles, and vasculonervous system of the palm of the hand. The longitudinal fibres, while playing the main role of anchoring system, have the freedom of movement of 1–2 mm in an anteroposterior direction, which is useful in the relative sliding of the structures during complex hand movements, such as grasping [1,2,3]. The transverse fibres, in their most distal component, perform the function of an additional pulley of the flexor tendons in the fingers, which constitutes a fibrous canal formed superficially by the proximal transverse ligament of the aponeurosis, deeply located by the palmar plate of the metacarpal bones and laterally by the septa of Legueu and Juvara. The vertical fibres are volumetrically the thinnest of the palmar fascia, with two main distribution sectors, namely the central compartment of the hand and the space of the septal canals, delineated by the eight deep vertical septa of Legueu and Juvara.

Dupuytren’s disease is the main pathology of palmar aponeurosis [1,2,4]. It is a chronic and progressive fibroproliferative lesion affecting most of the structures of the palmar and digital fascia and, less frequently, the thenar and hypothenar fascia. The fibrosis that characterizes this disease leads clinically first to the formation of skin retractions, followed by nodules and fibrous cords and, subsequently, to a forced contracture of the fingers and hand, responsible for the characteristic flexion deformity of the fingers, predominantly towards the fourth and the fifth rays [1,2,5,6].

Most of the therapies for Dupuytren’s disease are pharmacological therapy and aponeurotomy [7,8,9]. Collagenase is injected into Dupuytren’s diseased hands in order to dissolve the retracting fibrous cords and to induce a forced extension of the fingers to regain full mobility. It represents a palliative attempt which can allow for a delay in the operating room for younger patients. The selective aponeurotomy involves excision of only the diseased aponeurosis with the removal of the fascia in its longitudinal components and the sparing of the transverse fibres of Skoog. Conversely, the vertical septa of Legueu and Juvara are often not cut at their insertion base, due to the surgical criticality of the area. The excision of the diseased aponeurosis continues up to and including the natatory ligament and extends towards the digital space in cases where the disease has also expanded digitally, manifesting itself with retracting digital cords. This operation has a recurrence rate at 5 years of 20–40% [8]. In our opinion, the main cause of recurrence of the pathology is the limited knowledge of the three-dimensional anatomical structure of the fascia and the mechanism of pathophysiological evolution of the disease. To date, it is recognized that, during the pathology, the inflammatory molecules influence the production of collagen, with a consequent change in the extracellular matrix arrangement driving contraction and profibrotic signalling [10]. Moreover, it is well known that in Dupuytren’s disease, there is an increased amount of free nerve endings and nociceptors and this pathological innervation causes an increased secretion of nerve growth factors, which amplify fibrosis and play a role in pain onset [11,12].

Despite this, many aspects of the pathology have yet to be clarified. The aim of this study was to further deepen the knowledge on the anatomy of the palmar aponeurosis in Dupuytren’s disease, with particular attention to the microanatomical and histopathological differences between the longitudinal and vertical fibres. This can permit the implementation of the understanding of the pathophysiological mechanisms in Dupuytren’s disease, for innovative clinical implications and to guide surgeons in their decision on whether to remove the vertical septa.

## 2. Results

### 2.1. Immunohistochemistry and Morphometric Analysis of Vascularization

The haematoxylin-eosin stain showed the perpendicular organization of the vertical (V) and longitudinal (L) fibres of the palmar aponeurosis (Figure 1A). Figure 1B,C show the pattern of the fibres in the longitudinal (Figure 1B) and vertical (Figure 1C) portions, allowing us to distinguish areas of dense connective tissue (D), surrounded by loose connective tissue (L). The same organization is shown in the macroscopic histological image of the hand section, collected from the autopsy (Figure 1D), in which are evident: the vertical fibres, in continuity with the retinacula cutis, the transversal fibres, and the longitudinal ones, perpendicularly oriented. The basic anatomical features and the organization of the palmar fascia do not show any evident differences between the samples collected from Dupuytren’s disease and the autopsies.

The immunohistochemistry with Anti-Human Von Willebrand Factor antibody highlighted the endothelial cells (Figure 2A,B), allowing us to identify the arterial and venous components of the tissue. After counting and normalizing for the area (mm^2^), an average of 8.5 vessels/mm^2^ in the longitudinal fibres and generally more abundantly vascularized vertical fibres (24.6 vessels/mm^2^) were revealed (Figure 2C).

The anti-αSMA antibody for smooth muscle cells highlighted the arteries (Figure 3). The count analysis (Figure 4) showed an almost constant density, with the trend only slightly decreasing for the number of arteries in the longitudinal fibres as follows: 7.2/mm^2^ in the early stage, 5.1/mm^2^ in stage 2, and 3.3/mm^2^ in the advanced stage of the pathology. Conversely, the density of arteries in the vertical fibres showed a notable and progressive increase with the pathology’s evolution, where the mean value of the arteries was equal to 8.8/mm^2^ in the early stage, 5.2/mm^2^ in stage 1, 14.8/mm^2^ in stage 2, reaching a density of 21.4/mm^2^ in the advanced stages (Figure 4).

Anti-D2-40 immunostaining highlighted the presence of lymphatic vessels in both the longitudinal and vertical fibres of the palmar aponeurosis (Figure 5). The quantitative analysis demonstrated a progressively decreasing trend from the early stages of the pathology towards the advanced stages, in both longitudinal and vertical fibres, with a higher mean density of lymphatic vessels in the vertical sections (Figure 6). In detail, in the early stage of Dupuytren’s disease, the longitudinal fibres showed a mean density of lymphatic vessels equal to 2.5/mm^2^, which decreased to 1.1/mm^2^ in the stage 3–4. The density in the vertical septa then went, respectively, from 6.2/mm^2^ to 2.8/mm^2^.

The morphometric analysis showed a clear majority of vascularization in the collagen fibre area with respect to the adipose tissue, in both the longitudinal and vertical fibres, at both stages 0 and 4 of the Dupuytren’s disease (Table 1). With the evolution of the pathology, the distribution of arteries and lymphatic vessels was almost exclusively around the fibres and not in the adipose area.

### 2.2. Collagen Fibres Amount and Distribution

The immunostaining with anti-Collagen-I and III showed a homogeneous distribution and arrangement of the collagen fibres in both the longitudinal and vertical fibres (Figure 7). The collagen assay kit and the immunoblotting permitted a quantification of the collagen in the vertical and longitudinal septa at various stages of Dupuytren’s disease.

Figure 8 showed the µg of total collagen per mg of wet starting tissue in both the longitudinal and vertical fibres at all the stages of the pathology (from the early stages to stage 4). In the early stage, the quantity of collagen was equal to 33.4 µg/mg of tissue in the longitudinal fibres and 17.2 µg/mg in the vertical ones. In both the fibres, the collagen amount increased with the progression of the disease as follows: the values rose to ~55 µg/mg (stage 2) and ~51 µg/mg (late stage 3/4) in the longitudinal fibres, and up to ~46 µg/mg (stage 1) and then ~83 µg/mg (stage 2 and stage 3/4) in the vertical fibres (Figure 8). The increasing trend was greater in the vertical fibres, which showed a higher quantity of total collagen in each stage of the disease, with the exception of the early stage, compared to the corresponding longitudinal ones. Furthermore, in both the fibres, the increasing trend of collagen production was constant up to stage 2, and then settled down in the most advanced stage of the pathology (Figure 8).

The immunoblotting demonstrated a regular increase in the amount of Collagen-I with the progression of the pathology, with a similar trend in the longitudinal fibres (from 0.34 to 0.56, normalized values) and in the septa of Legueu and Juvara (from 0.27 to 0.64) (Figure 9A). In parallel, the Collagen-III amount decreased, such that the rich production of the early stages (normalized values equal to 0.74 in the longitudinal fibres and 0.63 in the vertical ones) gradually decreased in the more advanced stages (0.27 and 0.16, respectively) (Figure 9B). The data obtained from the control cadavers were 0.20 ± 0.01 and 0.19 ± 0.02, respectively, for Collagen-I and Collagen-III in the longitudinal fibres; and 0.14 ± 0.01 and 0.18 ± 0.001 in the vertical fibres, showing no statistically significant variations in the control samples between longitudinal and vertical fibres. In general, both Collagen-I and III showed a lower amount in the controls with respect to the pathological samples.

### 2.3. Inflammatory Markers

The inflammatory component of the pathological tissue was analysed to highlight a possible inflammatory process in the progression of the disease and identify any possible difference between the vertical septa and longitudinal fibres.

The immunohistochemical analysis using the anti-CD68 antibody labelled a rich abundance of mast cells (with blue-purple metachromatic-stained cytoplasmic granules, evident with Toluidine Blue 0.1%) and monocytes, fundamental key players in triggering the inflammatory context at an early stage of pathology in both the vertical (Figure 10A) and longitudinal (Figure 10B) fibres. The specificity of the immunostaining was demonstrated with the omission of the primary antibody (Figure 10C). However, the presence of monocytes and mast cells completely disappeared with the pathology’s progression (Figure 10D).

Immunoblotting with anti-IL-1β demonstrated (Figure 11B) a synergic behaviour between the vertical and the longitudinal fibres, where the normalized value of IL-1β was equal to 0.015 in the longitudinal fibres and 0.042 in the vertical ones in the early stages. Then, an increase in IL1β levels was observed in the intermediate stage 2 (values of 0.076 and 0.1 for the longitudinal and vertical fibres, respectively). Lastly, the inflammation levels decreased to the starting levels (0.054 and 0.032) in the advanced pathology.

The same trend was revealed by an ELISA immunoassay for the TNF-α protein, where the levels remained almost the same until stage 1 of the pathology (244 ± 82 pg/mg in the early stage, 321 ± 114 pg/mg in the stage 1, vertical fibres), then significantly increased (up to 1263 ± 364 pg/mg) in the intermediate stage 2, and finally returned to the initial levels (587 ± 97 pg/mg) (Figure 12). The same trend was revealed in the longitudinal fibres, with a maximum TNF-α protein level of equal to 950 ± 134 pg/mg at stage 2 of the pathology (Figure 12).

## 3. Discussion

Dupuytren’s disease is a progressive and insidious pathology characterized by fibromatosis and tissue retraction in the palm of the hand, and a gradual digital flexion [3]. The estimated 20–40% recurrence rate at five years after selective aponeurotomy surgery [8] remains unresolved and unexplained. An in-depth knowledge of the sophisticated three-dimensional structure of the palmar aponeurosis and the relative modifications during the progression of the pathology represents the indispensable prerequisite for understanding its functional role in the hand and the pathophysiological mechanisms of the Dupuytren’s diseases. For years, it was believed that Dupuytren’s disease exclusively involves the longitudinal fibres of the palmar aponeurosis, completely sparing the transverse and vertical fibres. However, the septa of Legueu and Juvara are in close anatomical relationship with the longitudinal fibres. This study confirmed the pathological changes in the longitudinal fibres with an increase in collagen production [13,14,15], but it also demonstrated, for the first time, the significant and often large biochemical involvement of the vertical fibres in the pathology. Moreover, the aim of this study was to analyse the progression of the pathology, thanks to the collection of samples from patients at different stages of the Tubiana’s classification [3].

No studies relating to vascularization in pathological longitudinal and vertical septa have been published in the literature. Shchudlo and coauthors (2018) investigated the structural and functional characteristics of palmar hypodermal tissue vascularization in Dupuytren’s diseased patients of different age groups, demonstrating that the younger group exhibited more severe constrictive remodelling of the palmar fascia in perforating the arteries supplying the hypodermis compared with the older patients, with more compensatory changes of its capillarization [16]. Holzer and coauthors (2013) demonstrated a higher expression of HIF-1α (hypoxia-inducible factor alfa) and VEGFR2 (vascular endothelial growth factor receptor 2), statistically significant compared to the controls, in the Dupuytren’s diseased tissue of palmar aponeurosis: the subsequent hypoxic conditions and angiogenesis are fundamental stimulators of fibrosis, indicating their participation in the myofibroblast-rich nodules as active disease processes [17]. In this study, it was highlighted as a strong vascular component in the diseased longitudinal fibres (8.5 ± 2.5 vessels/mm^2^), but even greater in the vertical fibres (24.6 ± 12.9 vessels/mm^2^) (Figure 2). Moreover, with the progression of the pathology, there was a corresponding increase in the arterial vessel density in the vertical fibres (from 8.8/mm^2^ in the early stage up to 21.4/mm^2^ in stage 3/4), while the longitudinal component maintained an almost constant arterial vascularization (under 7/mm^2^) in all the stages (Figure 4), with a homogenous distribution around the collagen fibres (Table 1). The increasing vascular supply in the vertical fibres correlated with a local inflammatory context (increase in IL-1β and TNF-α) (Figure 11 and Figure 12), in that the cytokine recruitment and neovascularization of the tissue are clear evidence that the septa of Legueu and Juvara are not spared from involvement in the pathology. In parallel, the lymphatic component showed a constant decrease, especially in the vertical fibres (from 6.2/mm^2^ to 2.8/mm^2^) (Figure 6), explaining that with the progression of the pathology and the fibrosis, an obstruction of the lymphatic vessels can occur, with a consequent decrease in drainage of fluids, along with the possible formation of oedema and progression of the inflammation process. The pathology starts with a prevalence of acute inflammation, as demonstrated by the abundance of monocytes and mast cells in the early stages of the pathology, and then by the increase in IL-1β and TNF-α, until stage 2 (Figure 10, Figure 11 and Figure 12). The acute inflammation then decreases, driving a fibrotic action [18], with the clear synthesis of new collagen (Figure 8 and Figure 9) and the typical clinical manifestations of the Dupuytren’s disease, such as the formation of nodules and cords and the tissue retraction.

The previous studies of the palmar aponeurosis in Dupuytren’s disease compared the altered tissue to healthy control, showing an increase in collagen-III and a parallel decrease in collagen-I in the diseased tissue [19,20,21]. Conversely, one limitation of this study was the absence of a healthy control tissue of palmar fascia from living patients, due to objective ethical difficulties in obtaining healthy material, as well as the collection of fewer samples in the early stages of pathology because patients only undergo surgery when they are in the more advanced stages of pathology. However, control samples from two cadavers were collected and examined for the basic anatomical characteristics and the collagen analysis. Other analyses from the cadaver samples were excluded due to the lack of reliability in comparing autopsies with living patients, due to protein degradation, and due to the normal deterioration process during the cadavers’ preservation process. The results showed that the starting normal values of Collagen III are lower, whereas the values of Collagen-I are similar with respect to the early stages of the pathology. These data confirmed the immediate increase in Collagen-III during the onset of the pathology, whereas Collagen-I increased in the advanced stages of the pathology.

The collagen values reported in the literature seems to contradict our results; however, they only made comparisons to the healthy tissue [22], and they did not evaluate the trend between the pathological stages, which is the new approach introduced in this study. Moreover, it is consistent to notice an initial increase in collagen-III in inflamed tissues, because this protein is strongly and rapidly produced in the inflammatory context [5,15,23] before it is progressively and constantly replaced by Collagen-I, a more stable and resistant molecule [24], which confers the typical fibrotic aspect of the pathological cords of the Dupuytren’s disease. The results of our immunoblotting confirmed the high production of Collagen-III in the early stages of the pathology, which progressively decreased in the more advanced stages (Figure 9), with an abundant production of collagen-I, typical of the pathological palmar aponeurosis. Our analysis demonstrated that the total collagen rose to ~51 µg/mg in the longitudinal fibres and ~83 µg/mg in the vertical fibres from the starting levels of 33.4 and 17.2 µg/mg, respectively (Figure 8). This significant increase in collagen demonstrated, for the first time, how both the longitudinal and vertical fibres of the palmar aponeurosis are actively involved in the Dupuytren’s disease. Moreover, these results highlight that the extracellular matrix behaviour is not stable during the progression of the disease, as a starting inflammatory process with the production of Collagen-III leaves room for further modifications of the matrix environment—the production of Collagen-I and a fibrosis condition. The increased production of collagen, together with expression of α-SMA by the fibroblast, are already recognized as key elements for the transformation of fibroblasts to myofibroblasts [21,25]. These conditions give the way to the clinical symptoms of the Dupuytren’s disease, which typically occur in the advanced stages of the pathology. So, to intervene on the patient in the best possible context, aiming to obtain good morpho-functional results and lower risk of surgical complications, it becomes fundamental to understand the staging of the pathology and adequate therapeutic timing.

The involvement of the vertical fibres demonstrated for the first time in this study, by specific analysis of vascularization, inflammation, and collagen deposition, could help to explain the high percentage of recurrences following selective aponeurotomy interventions. This therapeutic approach is, in fact, aimed at the excision of only the altered superficial palmar aponeurosis, completely sparing the healthy rays, the Skoog ligament, and the vertical septa. The surgical choice of excision regarding the vertical septa of Legueu and Juvara in the diseased rays could show a drastic reduction in relapses in patients. Future studies will aim to confirm these data by follow-up of the operated patients, to verify the main complications and recurrence rate of patients undergoing selective aponeurotomy with excision of the diseased vertical septa.

## 4. Materials and Methods

### 4.1. Sample Collection

A total of 12 samples were collected from patients affected by Dupuytren’s disease (mean age: 67 ± 8 years; 2 females, 10 males: 1 early stage, 1 stage 1, 6 stage 2, 4 stage 3/4, Tubiana’s classification), who had undergone selective aponeurotomy at the Plastic Surgery Unit of the University of Padua. According to the Tubiana’s classification, stage 0 (or early stage) indicates a palmar or digital nodule without deficit of extension and flexion contracture; stage 1 indicates an extension deficit between 0° and 45°; stage 2 between 45° and 90°; stage 3 between 90° and 135°; and stage 4 over 135° [3].

In accordance with the ethical standards regarding research on human tissues [26], a sample of the palmar aponeurosis, including a portion of longitudinal fibres and vertical septa (Figure 13), was sent to the Institute of Human Anatomy, Department of Neuroscience, University of Padua, for histological and biochemical analyses.

Each sample was divided into two pieces, where one portion was frozen at −80 °C after separation of the horizontal fibres from the vertical ones, for immunoblotting or collagen content analysis, while the second piece was fixed in 10% buffered formalin solution for histological and immunohistochemical analyses.

Fixed samples were dehydrated in graded ethanol and xylene, embedded in paraffin and cut into 5 µm thick sections by a microtome. Dewaxed sections underwent haematoxylin-eosin staining and immunohistochemical analysis.

Control samples of palmar fascia were collected from two freshly frozen cadavers (1 man, 77 y; 1 woman, 62 y) within the “Body Donation Program” of the Human Anatomy Section of the Department of Neurosciences, promoted by the University of Padova. The donors’ past medical history was free for tumours, hand trauma, osteotendinous disease and hand surgery. The specimens were examined by haematoxylin-eosin staining for the basic anatomical features, and both the longitudinal and vertical fibres were analysed for Collagen types I and III (by Western blot, as described below) to have a comparison range of healthy subjects, regarding collagen, which remains a rather stable protein even in autopsies.

### 4.2. Immunohistochemistry Staining

Dewaxed sections were treated with 2% H_2_O_2_ in PBS for 15 min to inhibit endogenous peroxidases and stained for Collagen I and Collagen III (for identification of collagen fibres), αSMA (α smooth muscle actin for the smooth muscle cells of the arteries), Von Willebrand Factor (for the identification of endothelial cells), D2-40 (for lymphatic vessels), and CD68 (for inflammatory cells).

The slides for Collagen I were treated using an incubation of EDTA (Ethylenediaminetetraacetate—Sigma Aldrich, St. Louis, MO, USA) pH 9, at 95 °C for 15 min, to permit the antigen retrieval, and were then washed in PBS. Following a 1 h incubation in blocking solution (PBS + 0.2% bovine serum albumin (BSA)), all the slides were incubated with the following primary antibodies overnight at 4 °C: Goat Anti-Collagen I (Southern-Biotech (Birmingham, AL, USA), 1:400); Mouse Anti-Collagen III (Abcam (Cambridge, UK), 1:300); Mouse Anti-Human Muscle Actin (Clone 1A4-Agilent Dako (Santa Clara, CA, USA), 1:200); Rabbit Anti-Human Von Willebrand Factor (vWF-Agilent Dako, 1:600); Monoclonal Mouse anti-D2-40 (1:100), Monoclonal Mouse Anti-human CD68 (Dako-Agilent, 1:3000). After repetitive PBS washing, the sections were incubated with the secondary antibodies conjugated with HRP (Horseradish Peroxidase) for 1 h in PBS + 0.2% BSA/Rabbit Anti-Goat (Jackson ImmunoResearch (West Grove, PA, USA), 1:300); Goat Anti-Mouse (Jackson ImmunoResearch, 1:500); Goat Anti-Rabbit (Jackson ImmunoResearch, 1:250). Negative controls underwent the omission of the primary antibody. The reaction was then developed with 3,3′-diaminobenzidine (Liquid DAB + substrate Chromogen System kit Dako), stopped with distilled water, and counterstained with Toluidine Blue 0.1%. All images were acquired by Leica DMR optical microscope (Leica Microsystem, Wetzlar, Germany).

### 4.3. Morphometric Analysis of Blood Vessels

Immunohistochemical analysis carried out with the Von Willebrand, α-SMA, and D2-40 markers were used to identify the presence of vessels (arteriovenous component, arterial vessels, and lymphatic vessels, respectively). Positive immunohistochemical reactions of the samples were analysed in serial sections (at least 2 slices for each sample) at 10× magnification. Computer-assisted image analysis using ImageJ software, version1.54j (freely available at http://rsb.info.nih.gov/ij/ (accessed on 16 June 2023)) was used to identify and count the vessels (with identification of vessels until 2 µm in diameter). It helped us to obtain the number per area and evaluate the percentage of vessels inside/between the collagen fibres compared to those located in the adipose tissue.

### 4.4. Western Blot

Immunoblotting quantified the collagen fibres (types I and III) and pro-inflammatory cytokine IL1β in the longitudinal and vertical septa of the palmar aponeurosis of the patients with different degrees of the pathology and in the control cadaver samples.

Briefly, the frozen samples were thawed, cut with a surgical scalpel, and mechanically digested; total proteins were extracted using RIPA lysis buffer (Thermo Scientific, Waltham, MA, USA) and quantified with the BCA Protein Assay Kit (Thermo Scientific). Equal amounts of proteins (30 µg) were separated on precast polyacrylamide gel with a gradient of 4–25% (Mini-PROTEAN^®^ TGX™ Precast Gels, Bio-Rad, Hercules, CA, USA) and transferred onto polyvinylidene difluoride membranes, PVDF (Bio-Rad). The membranes were incubated with a blocking solution for 1 h at room temperature, and then incubated overnight at 4 °C with the corresponding primary antibodies as follows:-Anti-Col III in rabbit (Abcam, 1:6000) in 0.5% BSA in PBS, blocking solution PBS + BSA 4%;-Anti-Col I in goat (Southern Biotech, 1:1000) in TBS (Tris-buffered saline solution) +5% non-fat dry milk, blocking solution TBS + 5% non-fat dry milk;-Anti-IL1β in goat (Biotechne (Minneapolis, MN, USA), R&D systems, 1:1000) in TBS + 3% BSA (Bovine Serum Albumin) + 0.1% Tween-20, blocking solution TBS + 3%BSA.

After repetitive washes in PBS or TBS, the membranes were incubated, respectively, with goat anti-rabbit–RP (Jackson ImmunoResearch, 1:5000) or rabbit anti-goat (Jackson ImmunoResearch, 1:12,000) antibody for 1 h at room temperature.

After more repetitive washes, the immunoreactive reaction was determined by SuperSignal™ West Pico PLUS Chemiluminescent Substrate (Thermo Scientific). The intensity of the bands was measured with the Top-Sensitivity Chemidoc Systems and Analysis Software (UVITEC, Cambridge, UK), and normalized on the same membrane based on the total protein amount transferred to the membrane, evaluated with Ponceau-S staining and subsequent analysis with ImageJ Software (Analyze Gel–Plot Lanes) [27]. Each protein (Collagen I, Collagen III and IL1β) was analysed in all the samples (both longitudinal and vertical fibres, from early stage to stage 3/4, and in controls) at least in duplicates.

### 4.5. Total Collagen Assay Kit

The Total Collagen Assay kit (ab222942, Abcam) was used to measure the collagen content in the longitudinal and vertical septa. Briefly, 50 mg of samples were cut into small fragments and homogenized using a mechanical homogenizer in distilled water (100 µL/10 mg). For each sample, 100 µL of the homogenate was added to 100 µL of NaOH 10N for 1 h at 120 °C. After cooling on ice, and then neutralization with 100 µL of HCl 10N, the samples were centrifugated at 10,000× *g* for 5 min. A total of 10 µL of each sample was transferred into 96 microwell plates and subjected to evaporation at 65 °C, until crystallisation. In each microwell, 100 µL of oxidation reagent mix (6 µL Chloramine T concentrate + 94 µL oxidation buffer) were added for 20 min at room temperature to allow for oxidation of the collagen molecules. For the development of the colorimetric reaction, 50 µL of developer solution (at 37 °C, 5 min), and then 50 µL of concentrate DMAB (*para*-Dimethylaminobenzaldehyde, 65 °C, 45 min) were added.

The absorbance values at 570 nm (by Victor-3 1420 Multilabel Counter, Perkin Elmer, Waltham, MA, USA) were converted into µg of collagen per mg of tissue, by the standard curve obtained with the Collagen standard solution (from 0 to 18 µg/mL).

### 4.6. ELISA Immunoassay

The human TNF-α ELISA kit (RAB1089, Millipore, Burlington, MA, USA) was used to quantify the levels of TNF-α in tissue lysates of longitudinal and vertical fibres of the palmar aponeurosis. Briefly, one sample for each stage of Dupuytren disease (early, 1, 2, 3/4) was randomly selected and analysed in triplicate. The standard curve was obtained with the Human TNF-α Protein Standard. For each sample, a minimum of 2 mg of protein per 1 mL of original lysate was used and diluted 10-fold with 1X Sample Diluent Buffer. A total of 100 µL of each sample were incubated in a Human TNF-alpha Antibody-coated ELISA plate overnight at 4 °C with gentle shaking. After 4 repetitive washes, 100 µL of Biotinylated detection antibody was added in each well for 1 h and then washed. After 45 min of incubation with 100 µL of the HRP-streptavidin solution at RT and subsequent washing, 100 µL of TMB substrate was added (30 min in the dark) to visualize the enzymatic reaction. Then, 50 µL of Stop Solution was used to stop the reaction and the OD values were immediately measured at 450 nm.

### 4.7. Statistical Analysis

Statistical differences between the longitudinal and vertical fibres at the different stages of the pathology for the amount of total collagen amount (µg/mg of tissue) and levels of TNF-α (pg per mg of total proteins) were tested by one-way analysis of variance, followed by Tukey’s test for multiple comparisons. The GraphPad Prism 3.0 statistical package (GraphPad Software Inc., San Diego, CA, USA) was used for the analysis. * *p* < 0.05; ** *p* < 0.01 were considered the limits for statistical significance.

## Figures and Tables

**Figure 1 ijms-25-06865-f001:**
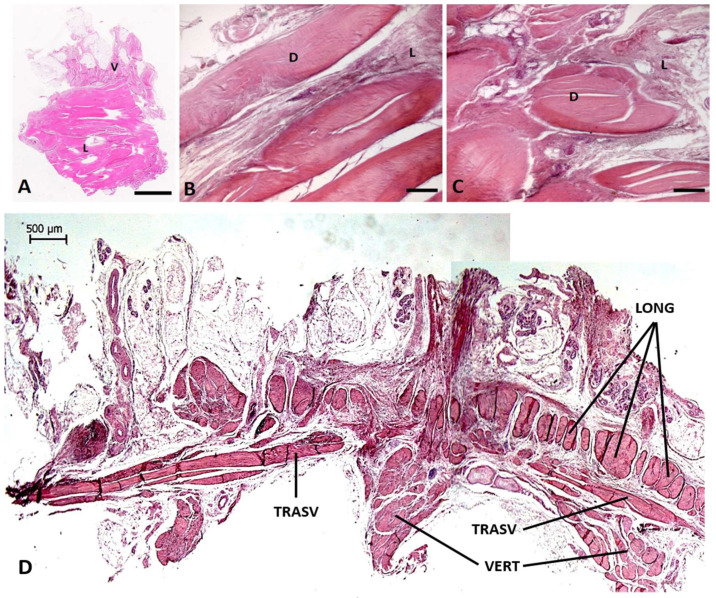
Haematoxylin-eosin staining of a section of palmar fascia from Dupuytren’s (**A**), in which are evident the longitudinal (L) and vertical (V) fibres. (**B**): Haematoxylin-eosin staining of the longitudinal fibres. (**C**): Haematoxylin-eosin staining of the vertical fibres. In (**B**,**C**) are shown areas of dense connective tissue (D) and loose connective tissue (L). (**D**): Macroscopic histological image of the hand section from autopsy, stained by haematoxylin-eosin. Long: longitudinal fibres of Wood Jones. Trasv: transversal fibres of Skoog. Vert: vertical septa of Legueu and Juvara. Scale bars: (**A**) = 2 mm; (**B**,**C**) = 200 µm; (**D**) = 500 µm.

**Figure 2 ijms-25-06865-f002:**
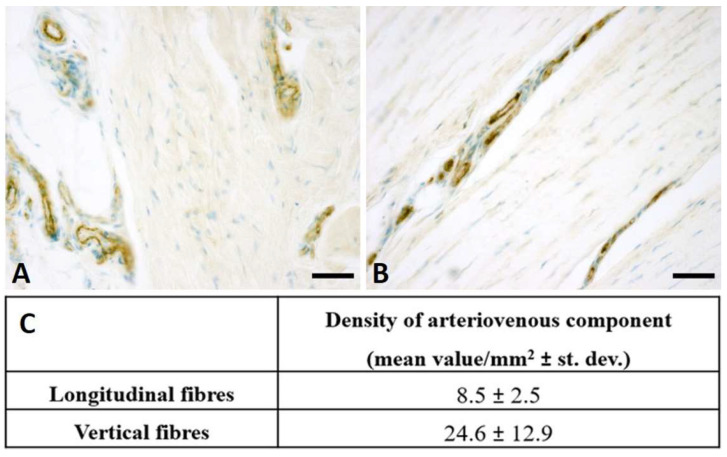
Anti-Von Willebrand Factor on vertical (**A**) and longitudinal (**B**) fibres, with positive endothelial cells. Scale bars: (**A**,**B**) = 50 µm. (**C**): Mean density of vessels obtained by anti-Von Willebrand Factor antibody stain, in longitudinal and vertical fibres of palmar aponeurosis of Dupuytren’s disease (mean values/mm^2^ ± standard deviation).

**Figure 3 ijms-25-06865-f003:**
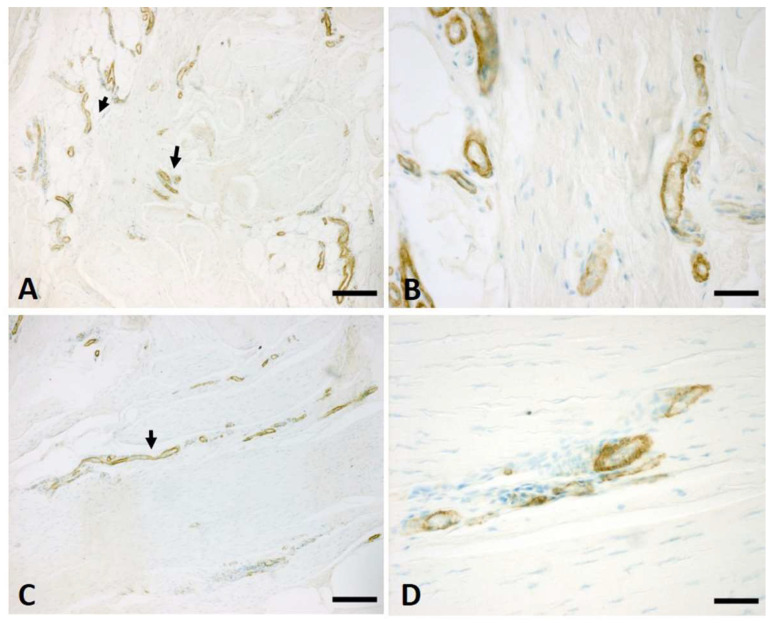
Anti-α-SMA on vertical (**A**,**B**) and longitudinal (**C**,**D**) fibres at stage 2 of the pathology. Arrows indicate arteries positive to anti-α-SMA antibody. Scale bars: (**A**,**C**) = 200 µm; (**B**,**D**) = 50 µm.

**Figure 4 ijms-25-06865-f004:**
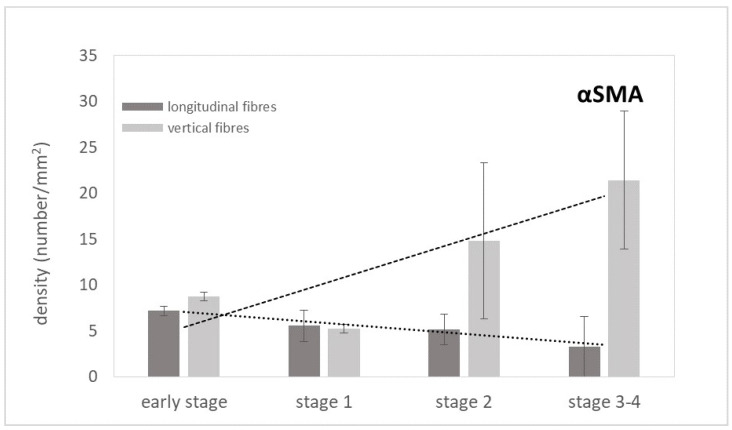
Mean density of arterial vessels and linear trend lines obtained by anti-α-SMA antibody stain in the longitudinal and vertical fibres of palmar aponeurosis of Dupuytren’s disease at early stage, stage 1, stage 2, and stage 3–4.

**Figure 5 ijms-25-06865-f005:**
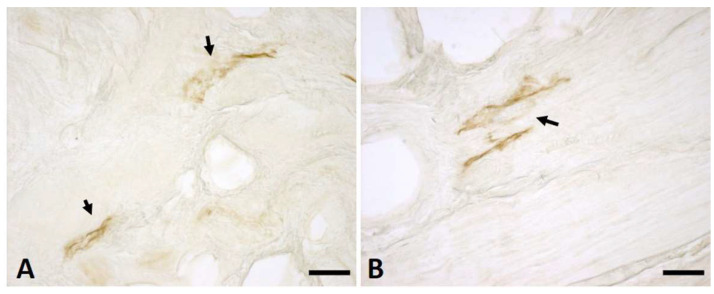
Anti-D2-40 on the vertical (**A**) and longitudinal (**B**) fibres. Arrows indicate positive lymphatic vessels. Scale bars: (**A**,**B**) = 50 µm.

**Figure 6 ijms-25-06865-f006:**
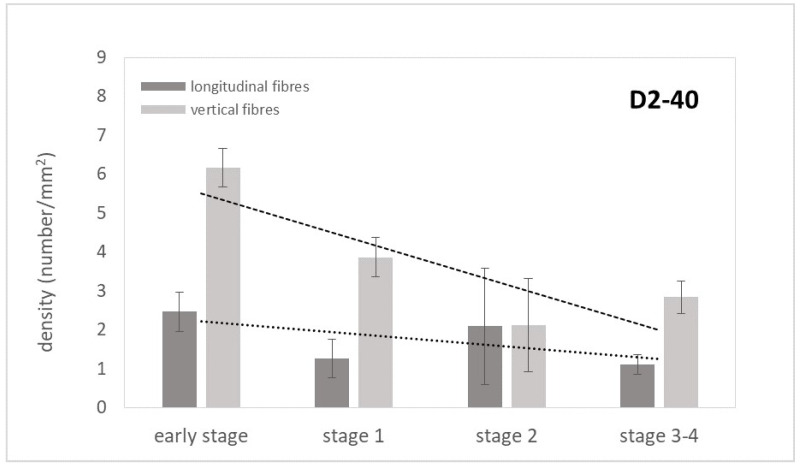
Mean density of lymphatic vessels and linear trend lines obtained by anti-D2-40 antibody in longitudinal and vertical fibres of palmar aponeurosis of Dupuytren’s disease at early stage, stage 1, stage 2, and stage 3–4.

**Figure 7 ijms-25-06865-f007:**
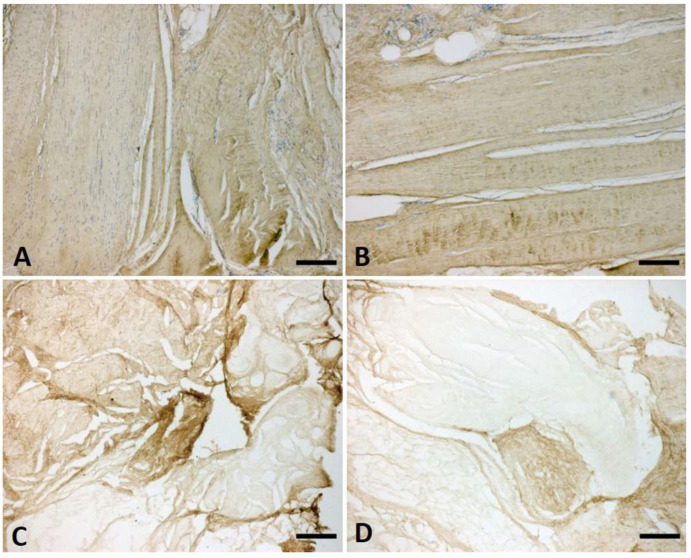
Anti-Collagen-I (**A**,**B**) and anti-Collagen-III (**C**,**D**) in vertical (**A**–**C**) and longitudinal (**B**–**D**) fibres of palmar aponeurosis at stage 2 of Dupuytren’s disease. Scale bars: 200 µm.

**Figure 8 ijms-25-06865-f008:**
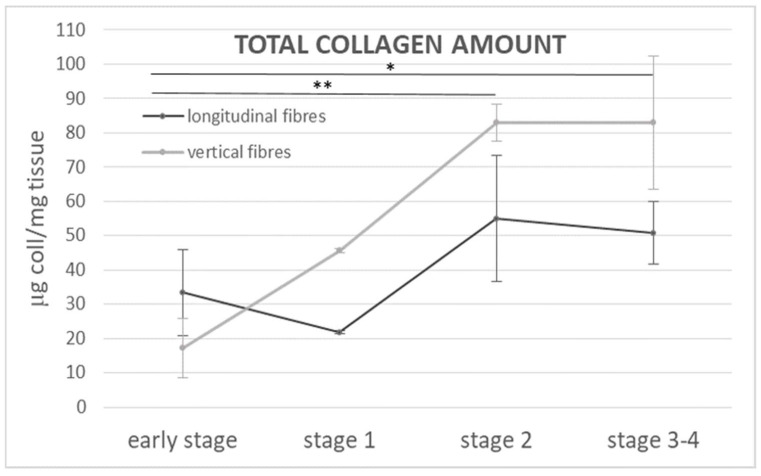
Total collagen amount (µg/mg if tissue) in the longitudinal and vertical fibres in the different stages of the pathology. Statistical differences between the four considered stages were tested by one-way analysis of variance, followed by Tukey’s test for multiple comparisons: * *p* < 0.05; ** *p* < 0.01.

**Figure 9 ijms-25-06865-f009:**
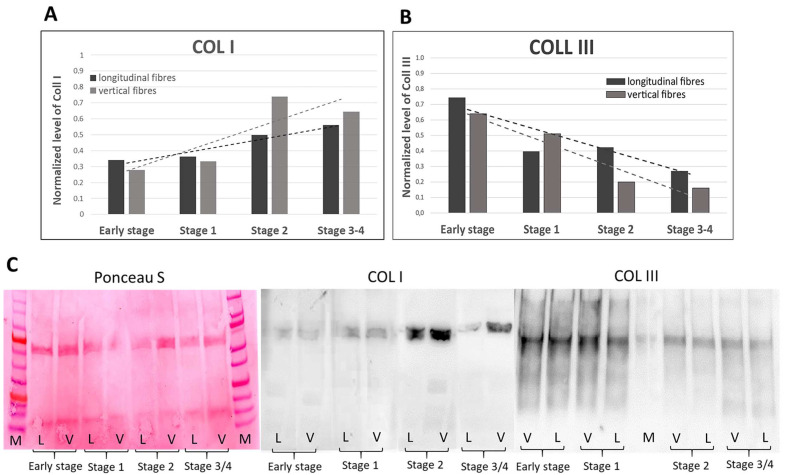
Normalized protein values and linear trend lines of Collagen I (**A**) and Collagen III (**B**) obtained by immunoblotting in the longitudinal and vertical fibres in the various stages of the Dupuytren’s disease (from early stage up to stage 3/4). (**C**) shows the chemiluminescent signal for Collagen I and Collagen III and one membrane stained with Ponceau S. M: marker; L: longitudinal fibres; V: vertical fibres.

**Figure 10 ijms-25-06865-f010:**
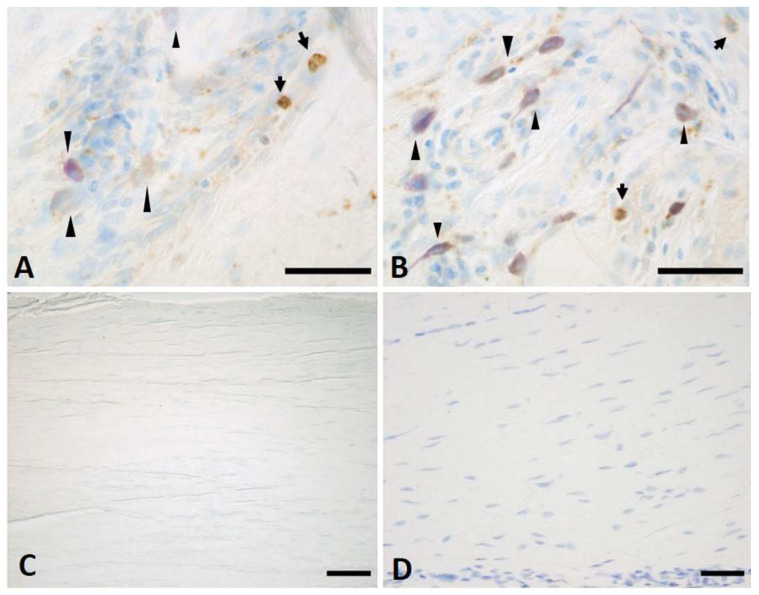
Anti-CD68 on vertical (**A**) and longitudinal (**B**) fibres at early stage of Dupuytren’s disease. (**C**) shows the negative control with the omission of the primary antibody. (**D**) is the immunostaining on longitudinal fibres at stage 2 of the pathology. Arrows indicate monocytes; arrowheads indicate mast cells. Scale bars: (**A**,**B**): 50 µm, (**C**,**D**): 100 µm.

**Figure 11 ijms-25-06865-f011:**
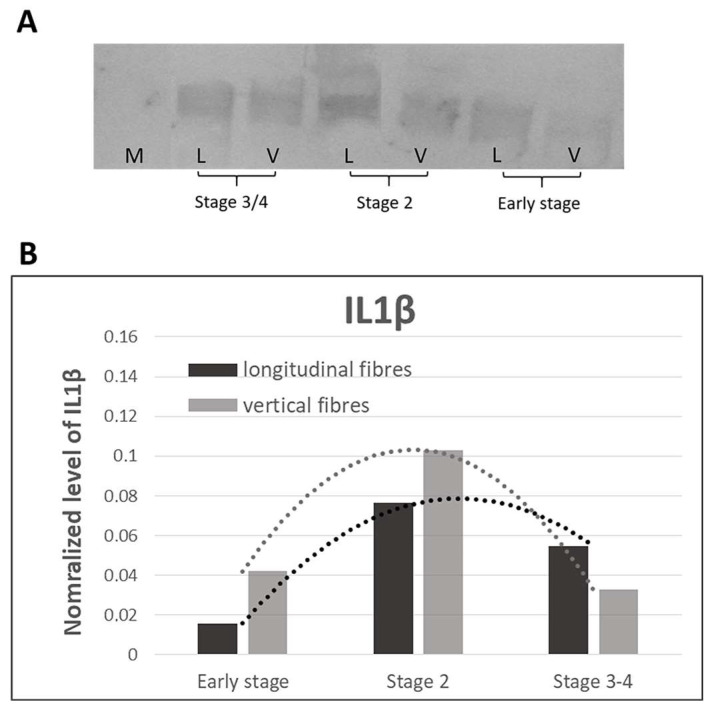
Chemiluminescent signal (**A**). (**B**) normalized protein values with relative trend lines of IL-1β obtained by immunoblotting in the longitudinal and vertical fibres at the various stages of the Dupuytren’s disease. M: marker; L: longitudinal fibres; V: vertical fibres.

**Figure 12 ijms-25-06865-f012:**
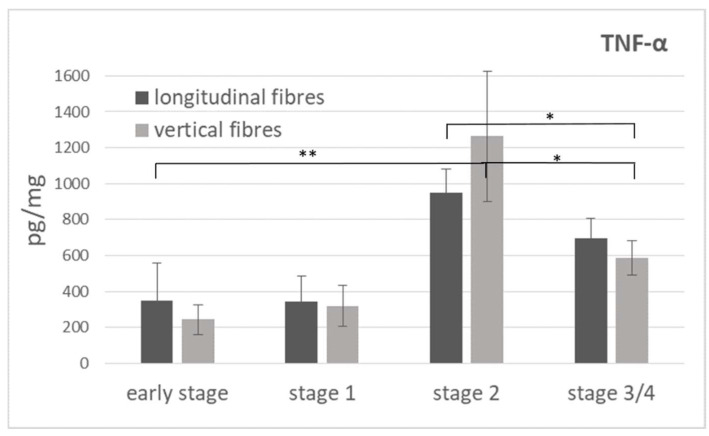
Levels of TNF-α (pg per mg of total proteins) obtained by ELISA immunoassay in tissue lysates of longitudinal and vertical fibres of the palmar aponeurosis from the early to advanced stages of Dupuytren’s disease. Statistical differences between the four considered stages were tested by one-way analysis of variance, followed by Tukey’s test for multiple comparisons: * *p* < 0.05; ** *p* < 0.01.

**Figure 13 ijms-25-06865-f013:**
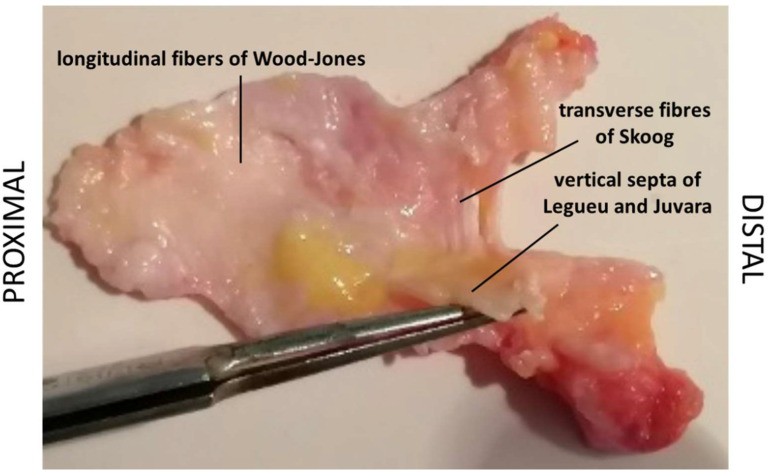
Sample of palmar aponeurosis of a patient affected by Dupuytren disease, observed in the dorsal–palmar direction: the deep surface of the longitudinal fibres is evident, with a physiological layer of loose connective tissue and adipose tissue adherent to the fibres. The longitudinal fibres continue towards the digital cords, partly visible in this picture, and towards the transverse fibres of Skoog and the vertical septa joined at the superficial origin and were sectioned at their deep end.

**Table 1 ijms-25-06865-t001:** Percentage of vascularization in the loose adipose tissue and between the collagen fibres, in the vertical and longitudinal areas, in early stage (stage 0) and advanced stage (stage 4) of Dupuytren’s disease, was calculated after stain with anti-Von Willebrand factor (arterial and venous components), anti-α-SMA (only arteries), and anti-D2-40 (lymphatic vessels).

		Von-Willebrand Factor		αSMA		D2-40	
		Stage 0	Stage 4	Stage 0	Stage 4	Stage 0	Stage 4
VERTICAL FIBRES	Adipose tissue	2.24%	5.80%	0.00%	3.50%	0.00%	0.00%
	Connective fibres	97.70%	94.20%	100.00%	96.50%	100.00%	100.00%
LONGITUDINAL FIBRES	Adipose tissue	7.70%	10.10%	0.00%	0.00%	0.00%	0.00%
	Connective fibres	92.30%	89.90%	100.00%	100.00%	100.00%	100.00%

## Data Availability

No new data were created or analyzed in this study. Data sharing is not applicable to this article.

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
