# Peer review of "Biochemical and Histological Differences between Longitudinal and Vertical Fibres of Dupuytren’s Palmar Aponeurosis and Innovative Clinical Implications"

_ijms, 2024, doi:10.3390/ijms25136865_

Round 1
Reviewer 1 Report
Comments and Suggestions for Authors
I read with great interest the paper by Fede et al. Overall the paper in interesting as dealing with an intriguing aspect of Dupuytren disease.
However, there are some major points that should be addressed:
With reference to the WB, in the material and methods section the authors stated that they “quantified collagen fibres (type I and III) and pro-inflammatory cytokine IL1β in longitudinal and vertical septa of the palmar aponeurosis of patients with 363 different degrees of the pathology.”
However, while data are reported in the graphs, the representative blots are not clear and incomplete. For instance, in figure 9 C, there are only two lines reported for both COL I and COL III, while the study groups are 4.
Again, in figure 11, there is only one WB line for cytokine IL1β reported. The authors should present the entire blot referring to IL1β expression in each experimental group, namely early stage, stage 2, stage 3-4.
Even the blots which have been uploaded by the authors in the supplementary material (original images for blots/Gels) are incomplete (since the blots from IL1β are missing), while the others are not clear; above each gel line it should be clearly indicated which group it refers to. Were data obtained on a single WB? Usually, gels are done at least in triplicate.
Another important point is the English language, which is not very fluent throughout the paper. Also, the anatomical description of the 3D network of longitudinal, transverse and vertical bundles of the palmar aponeurosis is not so easy to read and may be unclear for those who are not expert in the field. Therefore, in the opinion of the reviewer, the paper should be revised by a native English speaker.
Minor points:
- Line 92: Figures 2-B and 2-C; perhaps the authors meant figure 1; please correct.
- mm2 should be correct both in the abstract and results (more frequently in the figure legends)
- figure 10, figures should appear in the order they are described in the figure legend or vice versa, the figure legend should describe the figures in the order reported in the panels.
- Line 199: Arrows (↑) indicate monocytes; rhombus (â–²) indicate mast..; the symbol of the arrow is not necessary; please replace “rhombus” with arrowheads (again, the symbol of the triangle is not necessary).
Comments on the Quality of English Language
Moderate editing of English language required
Reviewer 2 Report
Comments and Suggestions for Authors
On one hand this is an interesting paper. It does explain the anatomy and fiber composition of palmar fascia, a structure most of us do not know very much.
One small thing: Word aponevrectomy does not exist, should be replaced with aponeurotomy.
One major problem has been identified: the lack of control, normal palmar fascia, wouldn’t it be possible to obtain 2 to 3 normal samples from autopsies? I think that this is important especially, that on one sample was obtained from an early stage and one from stage 1. This brings up another issue: what is the difference between early stage and stage 1? What is stage 0? A brief description or table on Tubiana classification would be helpful. To claim that fasciae from other sites are equivalent to palmar fascia is just a claim, not supported by any evidence. Their evidence is the similar content if collagen in various types of fasciae which does not indicate anything about histological and inflammatory changes and processes.
All the samples were obtained from older patients – that should make it easier to obtain normal tissues from autopsies as most would be done on older patients anyway. No controls are identified in any of the Figures.
Figure 2: A and C show low magnification of immunostained blood vessels, they do not contribute to the paper, should be omitted. We do not know anything about the presence of blood vessels in normal fascia because of the lack of control slides. It is not clear in Fig. 3 what pathology stage is represented.
Fig. 5: A and C show almost non-existent immunostaining, should be omitted. Fig. 9 shows immunoblot, or rather cut out lanes with staining, it is better to show entire lanes (this should be easy because they included the lanes in a supplement. Figure 10: not clear how the authors distinguished between monocytes and mast cells in these faintly stained sections. Figure 13: it really does not show the actual longitudinal fibers.
Though the paper is interesting the lack of controls is very concerning.
Comments on the Quality of English Language
just small think, aponevrectomy should be replaced by apneurotomy.
Round 2
Reviewer 1 Report
Comments and Suggestions for Authors
The authors have adequately addressed the comments raised by the reviewer. I have no further comments.
Comments on the Quality of English Language
minor revision
Author Response
Thank you to the Reviewer.
Reviewer 2 Report
Comments and Suggestions for Authors
It is disappointing that the authors refused to include any controls. Their explanation why they think that autopsy specimens would be inadequate because of protein degradation might be valid to some extent, they should at least compare basic anatomical features and distribution of collagens (those are pretty resilient. In addition, one does not need immunohistochemistry to identify blood vessels and the presence of inflammatory cells. To better understand evolution of Dupuytren it is important to compare with normal (granted, autopsy obtained) tissue with caveats pointed out in the manuscript..
Round 3
Reviewer 2 Report
Comments and Suggestions for Authors
I am glad that the authors were able to include proper controls.